# Identification of miRNA-Target Gene Pairs Responsive to *Fusarium* Wilt of Cucumber via an Integrated Analysis of miRNA and Transcriptome Profiles

**DOI:** 10.3390/biom11111620

**Published:** 2021-11-02

**Authors:** Jun Xu, Qianqian Xian, Ningyuan Zhang, Ke Wang, Xin Zhou, Yansong Li, Jingping Dong, Xuehao Chen

**Affiliations:** 1School of Horticulture and Plant Protection, Yangzhou University, Yangzhou 225009, China; 006963@yzu.edu.cn (J.X.); QQxian1996@163.com (Q.X.); ningyuan1214@foxmail.com (N.Z.); WK19825301461@163.com (K.W.); ZX957476929@163.com (X.Z.); liyansong202108@163.com (Y.L.); jpdong@yzu.edu.cn (J.D.); 2State Key Laboratory of Vegetable Germplasm Innovation, Tianjin 300192, China

**Keywords:** cucumber, miRNA, transcriptome, *Fusarium* wilt, resistance

## Abstract

*Fusarium* wilt (*FW*) of cucumber (*Cucumis sativus* L.) caused by *Fusarium oxysporum* f. sp. *cucumerinum* (Foc) is a destructive soil-borne disease that severely decreases cucumber yield and quality worldwide. MicroRNAs (miRNAs) are small non-coding RNAs (sRNAs) that are important for regulating host immunity because they affect target gene expression. However, the specific miRNAs and the miRNA/target gene crosstalk involved in cucumber resistance to *FW* remain unknown. In this study, we compared sRNA-seq and RNA-seq data for cucumber cultivar ‘Rijiecheng’, which is resistant to *FW*. The integrated analysis identified *FW*-responsive miRNAs and their target genes. On the basis of verified expression levels, we detected two highly expressed miRNAs with down-regulated expression in response to Foc. Moreover, an analysis of 21 target genes in cucumber inoculated with Foc indicated that *JRL3* (*Csa2G362470*), which is targeted by *miR319a*, and *BEE1* (*Csa1G024150*), *DAHP1* (*Csa2G369040*), and *PERK2* (*Csa4G642480*), which are targeted by *miR6300*, are expressed at high levels, but their expression is further up-regulated after Foc inoculation. These results imply that *miR319a*-*JRL3*, *miR6300*-*BEE1*, *miR6300*-*DAHP1* and *miR6300*-*PERK2* regulate cucumber defenses against *FW*, and provide the gene resources that may be useful for breeding programs focused on developing new cucumber varieties with enhanced resistance to *FW*.

## 1. Introduction

Cucumber (*Cucumis sativus* L.) is an important vegetable crop cultivated worldwide [1]. *Fusarium* wilt (*FW*) of cucumber, which is a typical and destructive soil-borne disease caused by *Fusarium oxysporum* f. sp. *cucumerinum* (Foc), is one of the major factors restricting global cucumber yield and quality [2,3,4,5].

During an infection, Foc easily penetrates cucumber plants and quickly spreads to the vascular tissues, where it occludes xylem vessels and produces a toxin that kills host cells, ultimately leading to *FW*. The disease results in the wilting of leaves or even the entire plant, with plant death occurring several days or weeks after infection [6,7]. *Fusarium* wilt is extremely difficult to control because it can occur during all cucumber growth stages [2]. Additionally, Foc can survive in the soil, straw or seeds for many years or even decades, leading to long-term disease cycles [8,9]. Furthermore, the changeable pathogenicity of Foc has limited the effectiveness of certain fungicides [3,10]. The most sustainable way to control *FW* involves the development and application of resistant cultivars. Thus, effective resistance genes need to be identified and the mechanism mediating disease resistance must be thoroughly characterized to develop *FW*-resistant cucumber cultivars.

Plant microRNAs (miRNAs) are endogenous small non-coding RNAs (sRNAs) that serve as crucial molecular regulators of the expression of functional genes by targeting mRNAs or by inhibiting translation [11,12,13]. Plant miRNAs participate in almost all biological processes, such as growth and development [14,15], hormone signal transduction [16] and responses to abiotic and biotic stresses (e.g., pathogen attack) [17,18]. For example, in rice, overexpression of *Osa*-*miR812w* increases resistance to *Magnaporthe oryzae*, whereas CRISPR/Cas9-mediated *miR812w* editing enhances disease susceptibility, suggesting that *miR812w* positively contributes to blast resistance [18]. In cotton, cleavage of *GhNAC100* mRNA by *ghr-miR164* decreases the amount of GhNAC100 that can bind to the CGTA-box of *GhPR3* promoter to repress expression, thereby enhancing the resistance to *Verticillium dahliae* [19]. Tomato *lncRNA23468* functions as a decoy RNA for *miR482b* to modulate the expression of *NBS-LRR* genes, resulting in increased resistance to *Phytophthora infestans* [20]. These results suggest that miRNAs mediate crucial plant defense gene regulatory pathways. Therefore, resistance-related miRNAs must be identified and their target genes associated with responses to pathogens should be functionally characterized. However, to date, very few miRNA-seq and RNA-seq analyses of cucumber defense responses to *FW* have been conducted. Thus, the miRNAs associated with *FW* resistance and their underlying mechanisms in cucumber remain unknown.

On the basis of an earlier examination of germplasm resources inoculated with Foc, we had confirmed that the cucumber cultivar ‘Rijiecheng’ was high-resistant against Foc, and might possess the resistant genes performing Foc defense [21]. In this study, we performed the integrated miRNA-seq and RNA-seq analyses to identify the *FW*-responsive miRNAs and their target genes in the *FW*-resistant cucumber cultivar ‘Rijiecheng’. Through the miRNA-seq from the Rijiecheng roots infected with Foc, we identified that eleven differentially expressed miRNAs (DEMs) were related to *FW* resistance. We further found that two miRNAs (*miR319a* and *miR6300*) had high expression levels and were down-regulated expression affected by the inoculation with Foc. Followed the RNA-seq analysis of Rijiecheng infected with Foc, 21 corresponding target genes of *miR319a* and *miR6300* had adverse expression tendencies and were expressed differently in cucumber roots inoculated with Foc. Based on the expression verification, we determined that *JRL3* (*Csa2G362470*), which is putatively targeted by *miR319a*, and *BEE1* (*Csa1G024150*), *DAHP1* (*Csa2G369040*), and *PERK2* (*Csa4G642480*), which are putatively targeted by *miR6300*, were abundantly expressed, with expression levels that were higher than those in the controls. These findings indicate that *miR319a*-*JRL3*, *miR6300*-*BEE1*, *miR6300*-*DAHP1* and *miR6300*-*PERK2* might be important for cucumber defenses against *FW.* Moreover, they may be useful for generating *FW*-resistant cucumber cultivars via breeding.

## 2. Materials and Methods

### 2.1. Plant Materials and Foc Treatment

This study was conducted using cucumber cultivar ‘Rijiecheng’, which was confirmed to be resistant to Foc during an earlier examination of germplasm resources inoculated with Foc [21]. Seeds were germinated on wet gauze in a Petri dish at 28 °C. The resulting seedlings were incubated in a growth chamber with the same conditions of 25 °C/18 °C cycle under a 16 h light/8 h dark cycle.

The Foc strain used in this study was isolated from cucumber roots exhibiting *FW* symptoms and then propagated on potato dextrose agar with different antibiotics inhibiting bacteria growth in plates at 28 °C for 4 days. The conidia were harvested and isolated for several times. Followed the inoculation phenotypes and sequencing verification, the Foc strain was confirmed and then cultured in potato dextrose broth in plates on a shaker (180 rpm) at 28 °C for 3 days. The concentration was adjusted to 10^6^ spores/mL using sterile distilled water prior to the inoculation of 14-day large cucumber seedlings at second true-leaf stage via a published dip-inoculation method [22]. Three replicates of seedling roots were harvested at 0, 24, 48, 96, and 192 h after the Foc inoculation. All samples were flash-frozen in liquid nitrogen and stored at −80 °C until analyzed.

### 2.2. Total RNA Isolation and Library Preparation for sRNA Sequencing

We collected three replicates of Foc-inoculated cucumber roots at different post-inoculation time-points (0, 48, and 96 h); non-inoculated roots were collected as the controls refer to the method as described by Dong and associates (2020) [22]. Total RNA for the real-time polymerase chain reaction (PCR) was extracted from flash-frozen cucumber roots using the MiniBEST Plant RNA Extraction Kit (TaKaRa, Dalian, China), after which RNA degradation and contamination were monitored on 2% agarose gels. Additionally, RNA purity was assessed using the NanoPhotometer^®^ spectrophotometer (IMPLEN, Calabasas, CA, USA), whereas the RNA concentration was determined using the Qubit^®^ RNA Assay Kit and the Qubit^®^ 2.0 Fluorometer (Life Technologies, Gaithersburg, MD, USA).

For each sample, 3 μg total RNA was used as the input material for constructing sRNA libraries. The sequencing libraries were generated using the NEBNext^®^ Multiplex Small RNA Library Prep Set for Illumina^®^ (New England Biolabs, Beverly, MA, USA) and index codes were added to attribute sequences to each sample. The sRNA sequencing analysis was completed using the AllPrep DNA-RNA-miRNA Universal kit (Qiagen, Duesseldorf, Germany), with DNA contaminants removed by an on-column DNase treatment.

### 2.3. miRNA-Seq Analysis

The miRNA sequencing analysis was performed by Novogene using the Illumina NextSeq 500 system. Expression levels were calculated as the number of transcripts per million (TPM). To identify the known miRNAs, the clean reads were used as queries for a BLAST search of the miRNA database miRbase 21.0 (http://www.mirbase.org/, accessed on 7 September 2021). The sRNA tags were mapped to the reference sequence using Bowtie (with no mismatches) to analyze their expression and distribution on the reference sequence [23]. We analyzed the miRNA families and their sequence conservation in miRbase, and the DEMs were further screened according to the following criterion: |log_2_FPKM (fold-change)| > 1 and *p* value < 0.05. For the unannotated sequences, we predicted new miRNAs using mireap (http://sourceforge.net/projects/mireap/, accessed on 7 September 2021). As miRNAs are primarily bound to the target site by complementary pairing, the data were analyzed using Miranda (http://www.microrna.org/microrna/home.do, accessed on 7 September 2021) to identify the targets of the mature miRNA sequences. The sRNA raw data were deposited in the NCBI Sequence Read Archive (accession number PRJNA760453).

### 2.4. MicroRNA Family Identification and Target Prediction

The miRNA families in other species were identified. In our analysis pipeline, known miRNAs were used along with miRNA.dat (http://www.mirbase.org/ftp.shtml, accessed on 7 September 2021) to search for families, whereas novel miRNA precursors were submitted to Rfam (http://rfam.xfam.org/, accessed on 7 September 2021) to screen for Rfam families. The miRNA expression levels were estimated as the number of TPM according to the following criteria [24]. The differential expression between two conditions/groups was analyzed using the DESeq R package (1.8.3). The *p*-values were adjusted according to the Benjamini and Hochberg method. A corrected *p*-value of 0.05 was set as the threshold for determining significant differences in expression. The target genes of miRNAs in plants were predicted using psRobot_tar in psRobot [25].

The TargetFinder software was used to predict miRNA target genes [26]. The candidate target genes of the DEMs were functionally characterized by a Gene Ontology (GO) enrichment analysis on the basis of the genome annotate of cucumber (http://cucurbitgenomics.org/organism/20, accessed on 7 September 2021).

### 2.5. Quantitative Real-Time RT-PCR (qRT-PCR) Assay of the miRNAs

The known and novel miRNAs were assayed by qPCR to validate relative expression patterns. The reverse transcription reaction was performed using the miRNA 1st Strand cDNA Synthesis Kit (by stem-loop) (Vazyme, Nanjing, China). The qPCR analysis was performed using the miRNA Universal SYBR^®^ qPCR Master Mix (Vazyme, Nanjing, China), with U6 snRNA used as the internal control. The miRNAs gene expression was calculated using qRT-PCR analysis, and expression data represent as the 2^−∆Ct^ method followed by further statistical analysis [27]. The standard deviation was measured for three biological replicates. The primers of the miRNAs used in the experiment were designed by miRNA Design V1.01 software and listed in Additional file 1: Appendix A.

### 2.6. Identification and Validation of Target Differentially Expressed Target Genes

We had analyzed the transcriptome of Foc-inoculated cucumber roots collected at different time-points after inoculation (0, 24, 48, 96, and 192 h). The generated data were submitted to the NCBI database (accession number PRJNA472169). The differentially expressed genes were considered with an adjusted FDR < 0.01 identified by DESeq and |log_2_FPKM (fold-change)| ≥ 1. The transcriptome was used to further screen for target genes of the miRNAs. Differences in the expression of the candidate genes were indicated by the color scale of the Toolbox for Biologists software.

The expression verification of the candidate genes was performed by the qRT-PCR analysis in the cucumber cultivar ‘Rijiecheng’ infected with Foc. The RNA samples of the Foc-inoculated cucumber roots were reverse transcribed into cDNA using the HiScript Q RT SuperMix for qPCR (Vazyme, Nanjing, China). The target gene-specific qRT-PCR primers were designed using the Beacon Designer 7.0 software. The cucumber tubulin alpha chain gene (*Csa4G000580*) was used as the internal reference control. The qRT-PCR analysis was performed using the Iqtm5 Multicolor qPCR detection system (Bio-Rad, Hercules, CA, USA) and the AceQ SYBR Green Master Mix (Vazyme, Nanjing, China), with three technical replicates per biological replicate. The expression data for three biological replicates were analyzed and are presented herein as the mean ± standard deviation. Primer information is provided in Additional file 1: Appendix A.

## 3. Results

### 3.1. Identification of miRNAs Responsive to Foc

To identify the miRNAs involved in plant responses to Foc, 15 sRNA libraries for three biological replicates of the Foc-inoculated roots at 48 and 96 h post-inoculation and the non-inoculated controls were prepared for a high-throughput sequencing analysis to identify miRNAs. We used the 18- to 24-nt sRNA sequences as queries to search for matches among the plant miRNA sequences in the miRBase 21.0 database. A total of 1185 miRNAs, including 1115 known miRNAs and 70 novel miRNAs, were identified in all the samples (Figure 1A). On the basis of detecting significant differences in expression in different groups using various methods, we further identified six differentially expressed miRNAs (DEMs) between the inoculated roots (R_48h_F) and the control (mock-inoculated) roots (R_48h_C) at 48 h post-inoculation, and seven DEMs between the inoculated roots (R_96h_F) and the control roots (R_96h_C) at 96 h post-inoculation, including two overlap DEMs in the two compare combinations (Figure 1B). The expression of these 11 independent miRNAs was significantly induced by Foc with the expression value from the different sequencing libraries visualized according to the changing colors (Figure 1C), suggesting these DEMs may be important for cucumber defense responses to *FW*.

### 3.2. Validation of miRNA Expression by qRT-PCR

To validate the identified DEMs, we completed a qRT-PCR assay to analyze the miRNA expression levels in the Foc-inoculated and control cucumber roots at three post-inoculation time-points (0, 48 and 96 h). The expression trends of the 11 selected miRNAs revealed by qRT-PCR were consistent with the sRNA sequencing data, and further reflected the reliability of the sRNA sequencing data obtained in this study (Figure 2). Among them, the five miRNAs contained the miR6300, miR398, miR398-3p miR319a and miR319a-3p had higher expression levels relative to other miRNAs and were obviously induced after inoculation with Foc. We aimed to use the *FW*-resistant cultivar ‘Rijiecheng’ to identify the resistant genes, and the accumulating evidence showed that miRNAs negatively regulate the corresponding target genes to defend against the disease infection [20]. Hence, we selected the *miR319a* and *miR6300* that had high expression value and were significantly down-regulated expression relative to the mock-inoculated controls, suggesting that they might regulate the increased expression of defense-related genes to enhance the resistance of cucumber cultivar ‘Rijiecheng’ to *FW*.

### 3.3. Identification of miRNA Target Genes and Profile Analyses

To further clarify the miRNA-mediated regulatory networks in cucumber inoculated with Foc, we predicted the target genes of miRNAs using psRobot_tar in psRobot [25]. The 1147 target genes identified for the 11 DEMs were subsequently analyzed on the basis of the transcriptome data for the ‘Rijiecheng’ samples inoculated with Foc. The *miR319a* and *miR6300* respectively possessed 51 and 501 target genes, and among them, 273 genes were further performed classification based on the functional analysis of the GO enrichment with the *p* value < 0.05 and were mainly classified in 14 groups (e.g., protein binding, protein kinase activity, transcription factors, oxidoreductase activity) associated with diversity functions (http://cucurbitgenomics.org/goenrich, accessed on 7 September 2021). However, the function characteristics of other genes were not annotated (Figure 3, Additional file 2: Appendix A). Based on the screening of differentially expressed genes via transcriptome of cucumber cultivar ‘Rijiecheng’ infected with Foc, we identified that 21 target genes were obviously affected by Foc. We also analyzed the association between *miR6300* and *miR319a* and their target genes via a functional characterization based on the bioinformatics method (Figure 4A, Additional file 3: Appendix A). Moreover, the expression of these 21 target genes was significantly up-regulated induced by Foc with the expression data from the different transcriptome libraries visualized according to the changing colors, in contrast to the miRNA expression trends (Figure 4B), indicative of a positive role for these up-regulated genes in cucumber cultivar ‘Rijiecheng’ defense responses to *FW*.

### 3.4. Verification of the Expression of Target Genes in Cucumber Infected with Foc

To confirm the Foc-induced expression of candidate target genes, the expression patterns revealed by the transcriptome analysis of cucumber roots infected with Foc were investigated by qRT-PCR analysis. We found that the expression of 21 target genes of two miRNAs were up-regulated after the Foc inoculation relative to the mock-inoculated controls, and these genes were highly or weakly expressed in the roots colored red and blue, respectively (Figure 5A). In detail, among them, the *JRL3* gene with a putative target site of *miR319a* as well as the *BEE1*, *DAHP1*, and *PERK2* genes with a putative target site of *miR6300* were better detected by a miRNA-target gene matching analysis. Additionally, these four genes had higher expression levels relative to the others and were significantly up-regulated affected by Foc at certain post-inoculation time-points, relative to the control levels (Figure 5B), indicating that these miRNA-target pairs might participate in cucumber defenses against *FW*.

## 4. Discussion

*Fusarium* wilt is a serious factor restricting global cucumber productivity [2,3]. The most sustainable method to control FW disease is the use of resistant cultivars to mine disease-resistance genes and better investigate disease-resistance mechanisms to develop FW-resistant cucumber cultivars.

Increasing research attention focus on the utility of functional miRNAs that directly regulate target genes for modulating host immune responses to enhance fungal disease resistance. For example, cotton *ghr-miR164* cleaves *GhNAC100* mRNA to regulate the expression of downstream disease-related genes and increase plant resistance to *Verticillium* wilt [19]. Additionally, in cotton, *miR5272a*-*GhMKK6*, *miR414c*-*GhFSD1* and *miR477*-*CBP60A* help regulate plant defenses against pathogens [28,29,30]. In potato, over-expression of *miR482e* results in the silencing of NBS-LRR protein-encoding genes and enhanced plant sensitivity to *V. dahliae* infections [31]. Other studies proved that *miR482b* negatively regulates tomato resistance to *P. infestans* [20,32]. In the model plant *Arabidopsis thaliana*, many miRNA regulatory networks, including those involving *miR156*-*SPL9*, *miR396*-*GRF*, *miR400*-*PPR*, *miR472*-*RDR6*, *miR773*-*MET2*, *miR844*-*CDS3* and *miR858*-*MYB*, mediate the resistance to pathogens [33,34,35,36,37,38,39]. In rice, miRNAs with functions influencing blast resistance have been reported, including miRNAs that positively (*miR7695*, *miR160*, *miR398*, *miR162*, and *miR166k-166h*) and negatively (*miR156*, *miR164*, *miR167*, *miR168*, *miR169*, *miR319*, *miR396* and *miR1873*) regulate immune responses [17,40,41,42,43,44,45,46,47,48,49,50,51]. Many miRNAs or target genes have been applied to protect various transgenic plants against diseases. However, to date, there have been relatively few reports describing miRNAs or their regulatory networks associated with cucumber immunity to *FW*.

Based on an earlier examination of germplasm resources inoculated with Foc, we had determined that the cucumber cultivar ‘Rijiecheng’ was highly resistant against Foc and might possess the resistant genes performing Foc defense [21]. Hence, we performed a comparative analysis of miRNA and transcript profiles in cucumber cultivar ‘Rijiecheng’ roots inoculated with Foc, we wanted to identify the DEMs and their target genes related to the *FW* responsive to better investigate the resistant mechanism of the cultivar ‘Rijiecheng’. In detail, we identified 11 independent DEMs from the miRNA sequencing (Figure 1), we further analyzed the expression patterns of these miRNAs in the Rijiecheng roots infected with Foc (Figure 2). Accumulating evidence shows that miRNAs negatively regulate the corresponding target genes to defend against the disease infection [20,31,32]. Therefore, we selected the *miR319a* and *miR6300* that had high expression levels and were obviously down-regulated after Foc infection, suggesting that these two miRNAs might affect the up-regulated expression of their target genes in the cucumber cultivar Rijiecheng to defense *FW* disease. The target genes of *miR319a* and *miR6300* were further screened by the transcriptome of Rijiecheng roots infected with Foc, and found that 21 genes differentially expressed after Foc attack (Figure 4). We further verified the disease-related variations in miRNAs and target genes, and revealed that the expression levels of *miR319a* and *miR6300* and their target genes *JRL3*, *BEE1*, *DAHP1* and *PERK2* are obviously affected by Foc (Figure 2 and Figure 5). Furthermore, the regulatory networks or functions of these two modules will be further investigated with phenotypes by the cucumber transgenic method. The other up-regulated miRNAs and their target genes will be further verified in the *FW*-susceptible cultivars and better identify the susceptibility genes to analyze the molecular mechanism of susceptibility of cucumber against *FW*. In the future, the availability of efficient multiplex transgene-free systems using new genome editing tools will be performed to introduce the broad-spectrum *FW* resistance by targeting multiple susceptibility genes simultaneously.

The *miR319a*-*JRL3*, *miR6300*-*BEE1*, *miR6300*-*DAHP1* and *miR6300*-*PERK2* pairs were confirmed that were obviously induced by the Foc. Additionally, the accumulated evidence shows that *JRL3* contains three jacalin-like lectin domains, and additionally, the effects of *JRL3* expression on plant defenses against pathogens have been investigated in several species. For example, in wheat (*Triticum aestivum*), *TaJRLL1* encodes mannose-specific jacalin-like lectin domains and regulates the salicylic acid-dependent and jasmonic acid-dependent pathways during defense responses to the fungal pathogen *Fusarium graminearum* and the biotrophic fungal pathogen *Blumeria graminis* [52]. The *BEE1* gene (brassinosteroid enhanced expression 1) encodes a bHLH domain. In soybean (*Glycine max*), the expression of a bHLH transcription factor gene, *GmPIB1*, is significantly induced by *Phytophthora sojae*, which results in the repressed expression of *GmSPOD1* and enhanced reactive oxygen species production to increase plant resistance to *P. sojae* [53]. *DAHP1* encodes a 3-deoxy-D-arabino-heptulosonate 7-phosphate synthase and significantly induce by pathogenic *Pseudomonas syringae* strains [54]. Considered together, these findings indicate that *miR319a* and *miR6300* might affect *JRL3*, *BEE1*, *DAHP1* and *PERK2* expression accordingly to regulate the resistance of cucumber to *FW*. This information may be relevant for future studies conducted to elucidate the genetic basis of *FW* resistance, with implications for the breeding of *FW*-resistant cucumber cultivar.

## 5. Conclusions

Based on a comparative analysis of sRNA-seq and RNA-seq data from the *FW*-resistant cucumber cultivar ‘Rijiecheng’ inoculated with Foc, we identified the Foc-responsive miRNAs and their target genes. Among them, *miR319a* and *miR6300* were highly down-regulated expression by the inoculation with Foc, and on the basis of RNA-seq analysis, 21 corresponding target genes were screened in cucumber roots inoculated with Foc. We further determined that *JRL3*, which is putatively targeted by *miR319a*, and *BEE1*, *DAHP1* and *PERK2*, which are putatively targeted by *miR6300*, were abundantly expressed, with expression levels that were higher than those in the controls. These results revealed that *miR319a*-*JRL3*, *miR6300*-*BEE1*, *miR6300*-*DAHP1* and *miR6300*-*PERK2* pairs might be important for cucumber defenses against *FW*.

## Figures and Tables

**Figure 1 biomolecules-11-01620-f001:**
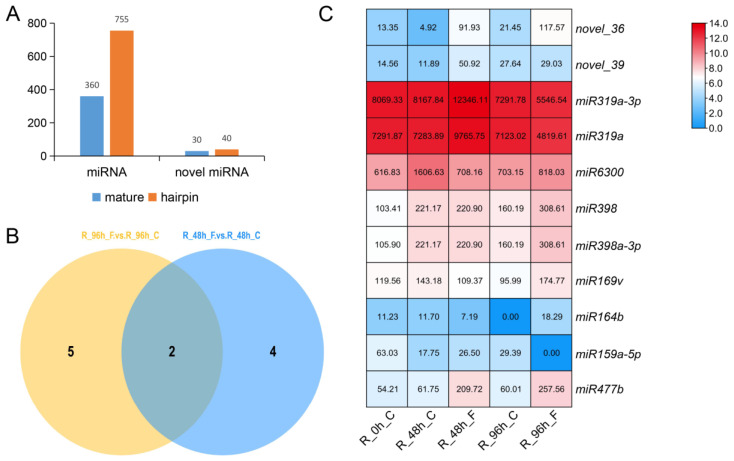
Identification of differentially expressed miRNAs (DEMs) responsive to *FW* in cucumber. (**A**) Distribution of known and novel cucumber miRNAs, including mature and hairpin miRNAs; (**B**) Venn diagram of the DEMs between the Foc-inoculated cucumber roots and the mock-inoculated control roots at 48 and 96 h post-inoculation; (**C**) Heatmap analysis of DEMs among different sequencing libraries. Red and blue indicate up-regulated and down-regulated expression, respectively.

**Figure 2 biomolecules-11-01620-f002:**
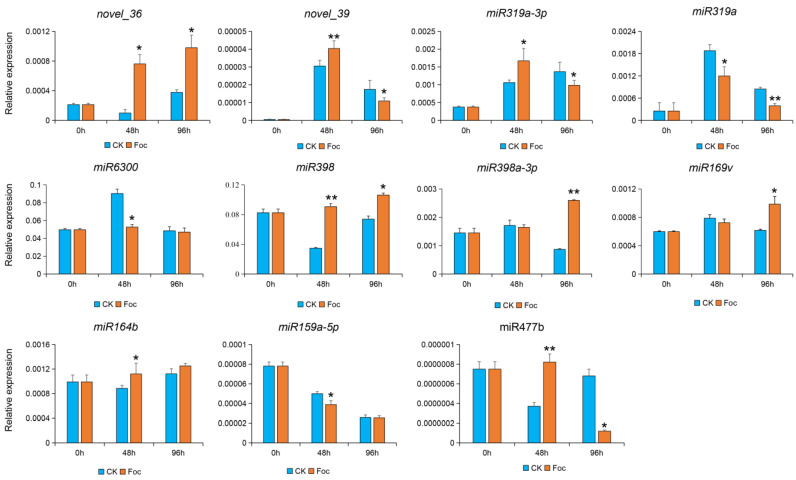
Validation of 11 DEMs in cucumber roots infected with Foc by stem-loop qPCR. Relative miRNA expression levels in the Foc-inoculated cucumber roots and the mock-inoculated control roots at 48 and 96 h post-inoculation were calculated according to the 2^−ΔCt^ method, with U6 snRNA used as the internal reference control. Data are presented as the mean ± standard deviation of three biological replicates. *: significantly different at *p* < 0.05; **: significantly different at *p <* 0.01.

**Figure 3 biomolecules-11-01620-f003:**
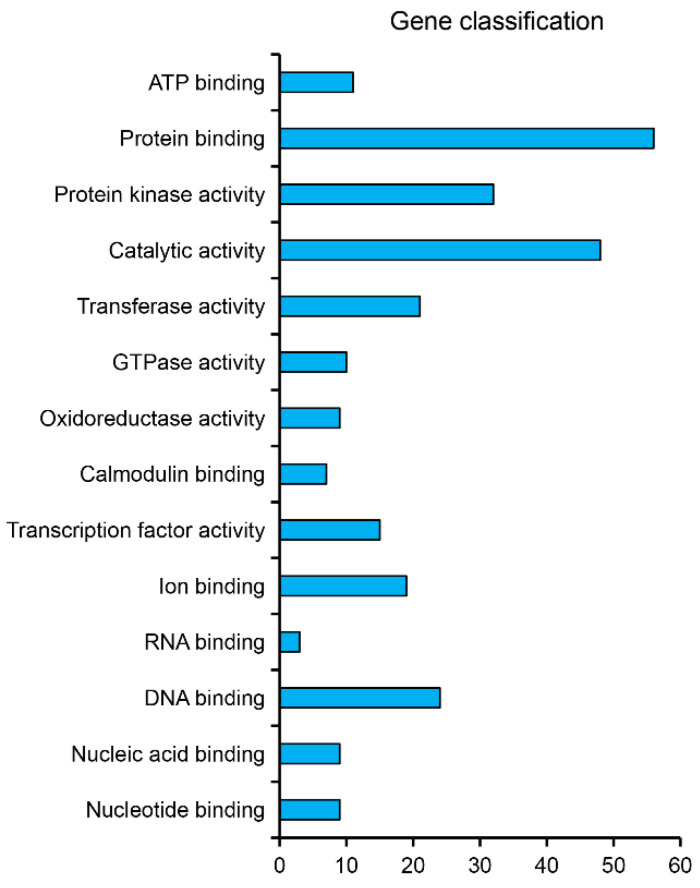
Functional classification of the targets of the *miR319a* and *miR6300*. The gene classification analysis was on the basis of GO enrichment. The ordinate represents the gene functions, and abscissa represents the number of genes.

**Figure 4 biomolecules-11-01620-f004:**
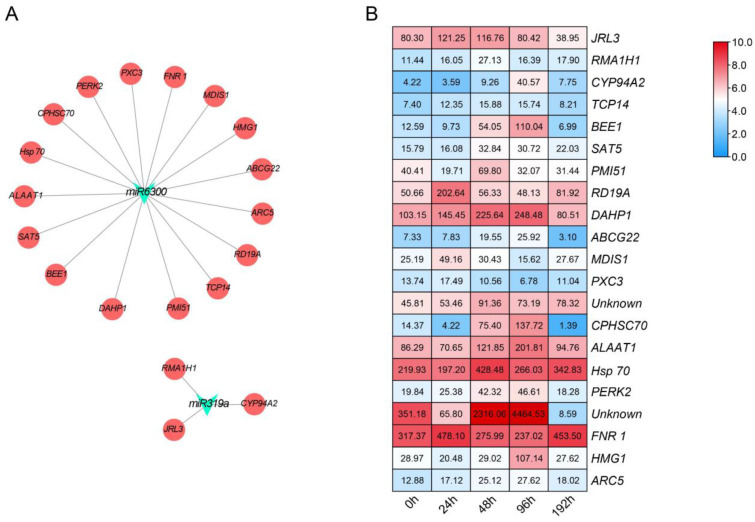
Target genes of selected miRNAs analyzed on the basis of the transcriptome of cucumber inoculated with Foc. (**A**) Predicted regulatory networks between *miR6300* and *miR319a* and the differentially expressed target genes; (**B**) Analysis of target gene expression profiles in cucumber roots after the Foc inoculation. ‘Rijiecheng’ roots inoculated with Foc collected at different post-inoculation time-points (0, 24, 48, 96 and 192 h) were used to analyze the target gene expression patterns. Genes more highly or more weakly expressed in the roots were colored red and blue, respectively. The RNA-seq data were submitted to the NCBI database (accession number PRJNA472169).

**Figure 5 biomolecules-11-01620-f005:**
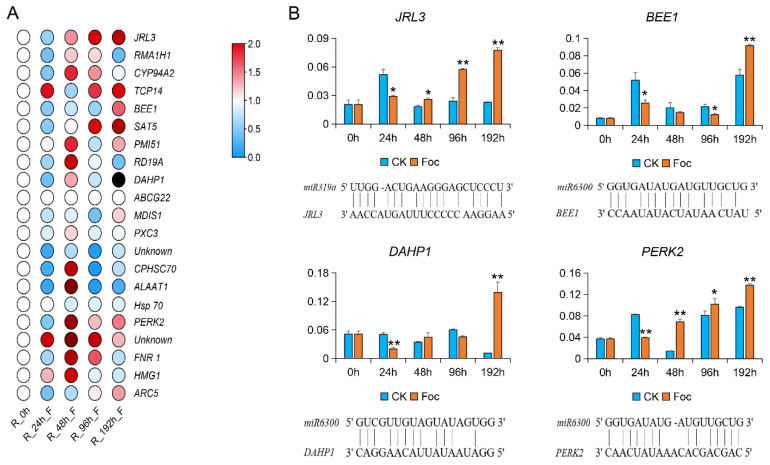
Validation of the target genes in cucumber inoculated with Foc. (**A**) Target gene expression levels in the Foc-inoculated cucumber roots and the mock-inoculated control roots were calculated according to the 2^−ΔΔCt^ method. Genes more highly or more weakly expressed in the roots were colored red and blue, respectively; (**B**) Expression patterns of four candidate target genes as well as the matches between *miR6300* and *miR319a* and selected target genes. Data are presented as the mean ± standard error of three biological replicates. *: significantly different at *p* < 0.05; **: significantly different at *p* < 0.01.

## Data Availability

The miRNA-Seq data associated with this study have been submitted to the NCBI SRA database (accession number PRJNA760453). Ethics approval and consent to participate, not applicable.

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
