# Peer review of "Identification of miRNA-Target Gene Pairs Responsive to Fusarium Wilt of Cucumber via an Integrated Analysis of miRNA and Transcriptome Profiles"

_biomolecules, 2021, doi:10.3390/biom11111620_

Round 1

Reviewer 1 Report

Plesa see the attached file

Author Response

Dear reviewer:

  Thank you so much for giving us an opportunity to revise our manuscript. We also greatly appreciate your valuable comments and constructive suggestions to improve the manuscript. According to your suggestions, we have tried our best to revise the manuscript and figures, and reorganize our results, which make the theme of paper clearer and read more smoothly. The detailed answers from your comments and suggestions are listed in the order.

We hope that you find this revised manuscript satisfactory for publication in Biomolecules journal. Thank you for your time and consideration.

We look forward to hearing from you.

Sincerely yours,

Xuehao Chen

Responses: (Q, question; A, answer)

  1. Q:But the Rfam link has to be updated : http://rfam.sanger.ac.uk is no longer valid.

A: Thank you very much for your careful examination. In the revised manuscript (line 136), we had amended the Rfam link as “http://rfam.xfam.org/”.

  1. Q:The usefulness of the informations given from line 133 to 141 are not clearly seen in the results part of the paper as the software and methods are not cited anymore..

A: Thank you so much for your comment. We had removed the usefulness information in this part. In the revised manuscript (line 143-146), the psRobot and TargetFinder softwares were used to predict the target genes of miRNAs in plants, and was performed in the result 3.3 to identify target genes of the miRNAs. The functional characteristics of the candidate target genes were performed to analyze and show in the Figure 3 and 4.

  1. Q: In the additional file 1 : Table S1, the primers listed from lines 71-82 clearly are not useful for this work.

A: Many thanks for your careful examination. We had removed the the usefulness primers in the Table S1.

  1. Q:The link between the 1.185 miRNAs and the 6+ 7 DEMs is not clear for me and need to be precised (lines 180-185).

A: Thank you so much for your suggestion. In the revised manuscript ( line 185 to 193), we had amended the descriptions to be precised. Briefly, we totally found 1,185 miRNAs in these samples, and among them, we further identified six differentially expressed miRNAs (DEMs) between the inoculated roots (R_48h_F) and the control (mock-inoculated) roots (R_48h_C) at 48 h post-inoculation, and seven DEMs between the inoculated roots (R_96h_F) and the control roots (R_96h_C) at 96 h post-inoculation based on the DESeq R package, the Benjamini and Hochberg methods.

  1. Q:Lines 185-186: the authors cannot write that « the expression of these 11 independent miRNAs was significantly induced by Foc (Fig. 1C) » as in the figure 1C, the heatmap shows a stability for 2 of them : miR319a-3p and miR319a

A: Thank you very much for your careful examinations. The high value of the miR319a-3p and miR319a caused the unobvious difference of the colour in the heatmap. In the revised manuscript ( line 193 to 195), following your comments, we had amended the Fig. 1C, and added the expression data of these 11 DEMs in accordance with p-value ≤ 0.01 and |log2FPKM (fold-change)| > 1 between the inoculated roots (R_48h_F and R_96h_F) and the control (mock-inoculated) roots to better visualize the expression difference of these miRNAs.

  1. Q: Lines 202-206 : The sentence is not very clear as the expression levels of miR6300 and miR319a are first described as high and then described as « down-regulated » ? These two miRNAs have clearly not the same expression profiles (figure 2C) but have both a lower expression level compared to the control.

A: Many thanks for your comments. We previously described the expression levels of DEMs using the 2−ΔΔCt method, and this figure showed the fold-change between the Foc-inoculated and the control (mock-inoculated) roots. In the revised manuscript ( line 210 to 219), we had amended Figure 2 using the 2−ΔCt method, and these results had better effects to visualize the high expression levels and the down-regulated tendency of the miR6300 and miR319a.

  1. Q: Line 219 : the authors wrote : «miR319a and miR6300 respectively possessed 51 and 501 target genes… » but in the additional file 3 : table S3 which is cited, only 273 genes are cited. Could they explain the link between the different values ?

A: Many thanks for your comments. The miR319a and miR6300 respectively possessed 51 and 501 target genes, however, 273 genes were further performed classification based on the functional analysis of the GO enrichment with the P value<0.05 (http://cucurbitgenomics.org/goenrich). In the revised manuscript (line 233 to 239), we added the detailed description to explain this question, and we also provided the information of the other 279 genes that were not identified by the GO enrichment in the Additional file 2: Table S2.

  1. Q: Lines 227-229 : the authors claim « the 21 target genes had obviously up-regulated expression levels » but this is absolutely not obvious for some genes in the related figure 4B.

A: Thank you so much for your comments. In the revised manuscript (line 239 to 248), we had amended the Fig. 4B, and added the expression data of these 21 differentially expressed genes (DEGs) from the transcriptome date in NCBI database accession number PRJNA472169. The high value of some genes in different time points caused unobvious difference of the colour in the heatmap, but had obviously up-regulated expression tendency.

  1. Q: Figure 4B : the meaning of the red and blue colors should be given.

A: Many thanks for your your careful examinations. In the revised manuscript (line 262 to 263), we had added the meaning of the red and blue colors described as “ Genes more highly or more weakly expressed in the roots were colored red and blue, respectively”.

  1. Q: Line 249 : in my opinion,considering the figure 5A, « mainly » is not appropriate in the sentence « The expression of 21 target genes of two miRNAs was mainly induced at 48 and 96 h after the Foc inoculation (Fig. 5A)»

A: Many thanks for your suggestion. In the revised manuscript (line 269 to 272), we had removed the « mainly », and amended the sentence to « We found that the expression of 21 target genes of two miRNAs were up-regulated after the Foc inoculation relative to the mock-inoculated controls, and these genes were highly or weakly expressed in the roots colored red and blue, respectively (Fig. 5A)».

Reviewer 2 Report

In this study, the authors identified two miRNAs which were down-regulated in response to Foc. In addition, the authors identified 21 putative target genes of these two miRNAs. The authors verified the expression pattern of these two miRNAs and 21 target genes by qPCR.

Line 48: delete both “the”.

Line 50: delete the

Line 53: delete the

Line 89: delete the

Line 203: miR6300 does not have higher expression level after inoculation with FW. Not sure if this is a typo or wrong label.

Major concern:

The authors simply analyzed two down-regulated miRNAs and ignore the other 9 miRNAs. The authors did not provide enough explanation to justify that. The authors need to at least discuss this point why they did not analyze the other 9 miRNAs and their putative target genes or why they are less important or not informative. Plant-pathogen interaction may simultaneously induce and suppress expression levels of certain genes which may contribute to resistance or susceptibility. However, the authors simply chose two miRNAs which were down-regulated and analyzed 21 of their target genes which were induced in response to Foc. The authors need to provide the information of other miRNAs and the analyses of their target genes or explain why they are not important to resistance to Foc.

Author Response

Dear reviewer:

  Thank you so much for giving us an opportunity to revise our manuscript. We also greatly appreciate your valuable comments and constructive suggestions to improve the manuscript. According to your suggestions, we have tried our best to revise the manuscript and figures, and reorganize our results, which make the theme of paper clearer and read more smoothly. The detailed answers from your comments and suggestions are listed in the order.

We hope that you find this revised manuscript satisfactory for publication in Biomolecules journal. Thank you for your time and consideration.

We look forward to hearing from you.

Sincerely yours,

Xuehao Chen

Responses: (Q, question; A, answer)

  1. Q:Line 48: delete both “the”, Line 50: delete the, Line 53: delete the, Line 89: delete the

A: Many thanks for your your careful examinations. In the revised manuscript, we had removed both “the”.

  1. Q:Line 203: miR6300 does not have higher expression level after inoculation with FW. Not sure if this is a typo or wrong label.

A: Thank you very much for your comments. We previously described the expression levels of DEMs using the 2−ΔΔCt method, and this figure showed the fold-change between the Foc-inoculated and the control (mock-inoculated) roots. In the revised manuscript (line 210 to 219), we had amended the Figure 2 using the 2−ΔCt method, and these results had better effects to visualize the high expression levels and the  down-regulated tendency of the miR6300 relative to the mock-inoculated controls.

  1. Q:The authors simply analyzed two down-regulated miRNAs and ignore the other 9 miRNAs. The authors did not provide enough explanation to justify that. The authors need to at least discuss this point why they did not analyze the other 9 miRNAs and their putative target genes or why they are less important or not informative. Plant-pathogen interaction may simultaneously induce and suppress expression levels of certain genes which may contribute to resistance or susceptibility. However, the authors simply chose two miRNAs which were down-regulated and analyzed 21 of their target genes which were induced in response to Foc. The authors need to provide the information of other miRNAs and the analyses of their target genes or explain why they are not important to resistance to Foc.

A: Many thanks for your comments and suggestions. In the revised manuscript, we added the amended figure, results, and corresponding discussions to explain the selective miRNAs and target genes. In detail, in the revised manuscript (line 210 to 219), we firstly amended the Figure 2 using the 2−ΔCt method, which had better effects to visualize the expression levels and tendency of the miRNAs relative to the controls. Among them, miR6300 and miR319a had higher expression level and were down-regulated after Foc infection, indicated that they might affect the target genes up-regulated expression in the cucumber cultivar ‘Rijiecheng’ to defense the Foc. Moreover, we provided the discussion (line 313 to 328) to explain the meaningful of the selective miRNAs. Briefly, we selected the cucumber cultivar ‘Rijiecheng’, which was high-resistant against the Foc and might possessed the up-regulated resistant genes after the Foc attack. Therefore, we performed the integrated analysis of miRNA and transcriptome profiles to identify candidate miRNAs and their target genes related to FW defense. Furthermore, we also added the descriptions (line 333 to 339) to discuss the meaningful of the other miRNAs with the up-regulated expression patterns and target genes in the future studies.

Reviewer 3 Report

The topic covered in this manuscript addresses an interesting topic for farmers and consumers alike. However, it lacks magnitude since, out of the multitude of existing cucumber varieties, only specimens belonging to the “Rijiecheng” cultivar are evaluated.

Structurally, the manuscript is balanced, with well-proportioned sections and relevant subchapters, and the subheadings are suggestive. Also, the charts are relevant, their quality is high and the explanations offered in the caption are sufficient and relevant.

The bibliography is up to date and relevant for the analyzed subject.

However, the quality of the manuscript needs to be significantly improved in some parts:

- Introduction - provides relevant information about the chosen topic, the bibliography is adequate, but the authors fail to clearly address the objectives of the study so that it is impossible for me to evaluate their fulfillment/achievement; from my point of view, the paragraph between lines 61-73 must be completely rewritten and the authors must clearly state the purpose/objectives of the research (null hypothesis H: 0).

- Materials and method - are briefly written but prove that the authors have a very good command of molecular manipulation techniques; however, some clarifications are needed:

line 81: Does the cycle involve day/light also? or only temperature?

line 83: Please provide the isolation of Fusarium stain from roots in detail.

line 87: What is the age of seedlings corresponding to the "second true-leaf stage"?

line 87-88: please refer to the method as ... [22]

line 92: in the methodology section please do not refer as "our previous ..results"

line 126: I am not sure if Rfam is available on indicated website

- The results - are judiciously presented and the data are relevant; here is one suggestion:

line 172-179: authors should stick to presenting results; elements of methodology should be provided in the upper section.

- Discussions - unfortunately, this section is the weakest part of the manuscript; the authors fail to critically evaluate their own results and fail to connect them with existing literature; the paragraph between lines 267-271 is redundant, these details were presented in the introduction; authors should remove this section.

- From my point of view, the authors have to completely rewrite the Discussion section in which to objectively discuss their own results

Congratulations to the authors on their efforts and for the interesting findings presented and hope my suggestions will help to improve the manuscript! 

Author Response

Dear reviewer:

  Thank you so much for giving us an opportunity to revise our manuscript. We also greatly appreciate your valuable comments and constructive suggestions to improve the manuscript. According to your suggestions, we have tried our best to revise the manuscript and figures, and reorganize our results, which make the theme of paper clearer and read more smoothly. The detailed answers from your comments and suggestions are listed in the order.

We hope that you find this revised manuscript satisfactory for publication in Biomolecules journal. Thank you for your time and consideration.

We look forward to hearing from you.

Sincerely yours,

Xuehao Chen

Responses: (Q, question; A, answer)

  1. Q:Introduction - provides relevant information about the chosen topic, the bibliography is adequate, but the authors fail to clearly address the objectives of the study so that it is impossible for me to evaluate their fulfillment/achievement; from my point of view, the paragraph between lines 61-73 must be completely rewritten and the authors must clearly state the purpose/objectives of the research (null hypothesis H: 0).

A: Many thanks for your comments and suggestions. In the revised manuscript (line 63 to 80), we had amended and rewritten the paragraph to better state the purpose/objectives of this research.

  1. Q:Materials and method - are briefly written but prove that the authors have a very good command of molecular manipulation techniques; however, some clarifications are needed:

A: Thank you so much for your suggestions. In the revised manuscript, we had amended and added the detailed descriptions of some methods.

  1. Q:line 81: Does the cycle involve day/light also? or only temperature?

A: Many thanks for your careful examinations. In the revised manuscript (line 88 to 89), we had added the conditions of the day/light cycle.

  1. Q:line 83: Please provide the isolation of Fusarium stain from roots in detail.

A: Many thanks for your suggestion. In the revised manuscript ( line 90 to 94), we had added the detailed description of the isolation of Fusarium stain.

  1. Q:line 87: What is the age of seedlings corresponding to the "second true-leaf stage"?

A: Many thanks for your comments. In the revised manuscript (line 96), we had amended the description of the age of seedlings.

  1. Q:line 87-88: please refer to the method as ... [22]

A: Many thanks for your comments. In the revised manuscript (line 102 to 104), we had amended the description.

  1. Q:line 92: in the methodology section please do not refer as "our previous ..results"

A: Many thanks for your suggestion. In the revised manuscript (line 102), we had amended and removed the descriptions.

  1. Q:line 126: I am not sure if Rfam is available on indicated website.

A: Many thanks for your comments. In the revised manuscript(line 136), we had amended the available website.

  1. Q:line 172-179: authors should stick to presenting results; elements of methodology should be provided in the upper section.

A: Many thanks for your suggestion. In the revised manuscript, we had amended and removed the elements of methodology in the results.

  1. Q:Discussions - unfortunately, this section is the weakest part of the manuscript; the authors fail to critically evaluate their own results and fail to connect them with existing literature; the paragraph between lines 267-271 is redundant, these details were presented in the introduction; authors should remove this section.

A: Many thanks for your suggestions. In the revised manuscript, we had removed some repeated descriptions, and rewritten the discussion to further explain our results and connected the innovative literature to better improve our manuscript.

Round 2

Reviewer 2 Report

This manuscript is better improved after the revision.

Reviewer 3 Report

I am satisfied with the corrections made and I consider that the manuscript has been improved. Congratulations to the authors!